# Prolonged preoperative wait time associated with elevated postoperative thirty-day mortality following intracranial tumor craniotomy in adult patients: A retrospective cohort study

Zhichao Gao[1], Yuhang Zhang[2], Jiaqing Guan[1], Weifeng Dong[1], Cheng Huang[3]*

1 Department of Neurosurgery, The First People's Hospital of Xiaoshan District, Hangzhou, Zhejiang, China, 2 Department of Orthopedics, The First People's Hospital of Xiaoshan District, Hangzhou, Zhejiang, China, 3 Departmentof Coloproctology, The First People's Hospital of Xiaoshan District, Hangzhou, Zhejiang, China

* drhuangcheng2021@163.com

## Abstract

### Objective

Prior studies have established preoperative wait time as a potential risk factor for postoperative outcomes across various clinical conditions. However, associations between wait time and short-term prognosis following intracranial tumor surgery are still largely unknown. Our study sought to investigate associations between pre-operative wait time and postoperative thirty-day mortality following intracranial tumor craniotomy in adult patients.

### Methods

This retrospective cohort study utilized data extracted from the ACS NSQIP data-base, comprising 18,298 adult patients who underwent intracranial tumor craniotomy between 2012 and 2015. The primary exposure and outcome were preoperative wait time and postoperative thirty-day mortality, respectively. Smooth curve fitting evaluated the linear or nonlinear association between them. The effects of exposure on outcome were evaluated using multivariate Cox proportional hazard regression models and Kaplan-Meier curves. Subgroup analyses and interaction testing were conducted to evaluate the effect modification of confounding factors. The robustness of the main results was assessed through propensity score matching and sensitivity analyses.

### Results

Prolonged preoperative wait time was independently and linearly related to elevated thirty-day mortality (HR = 1.075, 95%CI: 1.040–1.110). The ventilator-dependent status significantly modify the relationship between wait time and mortality. The linear wait time-mortality association was observed solely in non-ventilator-dependent

**Data availability statement:** All relevant data are within the paper and its Supporting Information files.

**Funding:** The author(s) received no specific funding for this work.

**Competing interests:** The authors have declared that no competing interests exist.

**Abbreviations:** ACS NSQIP, American College of Surgeons National Surgical Quality Improvement Program; BMI, body mass index; ASA, American Society of Anesthesiologists; CPT, Current Procedural Terminology; Na, serum sodium; BUN, blood urea nitrogen; WBC, white blood cell; HCT, hematocrit; INR, international normalized ratio; COPD, chronic obstructive pulmonary disease; CHF, congestive heart failure; SD, Standard deviation; HR, hazard ratio; CI, confidence interval; Ref, reference; STROBE, Strengthening the Reporting of Observational Studies in Epidemiology.

patients, showing an 8.3% increase in thirty-day mortality risk for each additional day of waiting (HR = 1.083, 95%CI: 1.049–1.119). Patients who waited ≥1 day had a 0.74% higher absolute risk and a 31.3% higher relative risk of thirty-day mortality compared to those who waited <1 day. The sensitivity analyses corroborated the robustness of these results.

## Conclusions

Prolonged preoperative wait time has an independent linear association with elevated postoperative thirty-day mortality in non-ventilator-dependent adult patients undergoing intracranial tumor craniotomy. Clinicians should minimize preoperative wait time to mitigate the risk of thirty-day mortality. Nonetheless, further research is warranted to validate the results and establish causality.

## Introduction

Intracranial tumors represent a significant health challenge, with global incidence rates of about 21 cases per 100,000 people, accounting for roughly 2% of all cancers in humans and contributing substantially to oncological morbidity and mortality worldwide [1,2]. Operative resection through craniotomy remains the major treatment for most intracranial tumors [3]. However, craniotomy also carries considerable perioperative risks such as postoperative hemorrhage, infection, epilepsy, and vascular events, as well as short-term mortality [4,5]. The thirty-day mortality is commonly used to evaluate surgical safety and risk [6,7]. The reported thirty-day mortality rates following intracranial tumor craniotomy ranged from 1.4% to 3.0% across various international cohorts [8–13]. These notable mortality rates underscore the imperative to identify modifiable perioperative factors that could inform risk stratification and optimize clinical outcomes.

Previous research has identified some perioperative factors associated with thirty-day mortality following intracranial tumor craniotomy. Patient-specific factors including age, functional health status, body mass index (BMI), and American Society of Anesthesiologists (ASA) classification consistently demonstrate prognostic value [7,14]. Tumor characteristics such as tumor metastasis significantly alter risk profiles compared to primary lesions [12]. Treatment-related variables reveal differential outcomes between two surgical procedures (biopsy vs. resection) [6], with intraoperative interventions like steroid administration modifying risk profiles [9]. Clinical and laboratory parameters exhibit stratified predictive capacity, encompassing both preoperative indicators (platelets [15], hematocrit [16], blood urea nitrogen [17], serum sodium [18]), and postoperative changes (hemoglobin drift [10], sepsis/septic shock [11], and respiratory complications [12]). Despite these advancements, contemporary risk models persistently overlook the potential impact of preoperative wait time—a clinically modifiable factor that may significantly influence patient outcomes.

Preoperative wait time, usually defined as the interval from diagnosis or hospital admission to surgical intervention, is considered a potential risk factor associated

with postoperative outcomes. Prolonged wait time may be associated with disease progression, physical deterioration, and increased psychological burden on patients [19]. Several studies in surgical oncology have demonstrated associations between prolonged preoperative wait time and poorer postoperative outcomes in multiple cancer types such as head and neck cancers, breast cancers, colorectal and colon cancers, and lung cancers [20–22]. In the field of neurosurgery, studies of lumbar disc herniation and cervical spondylotic myelopathy have demonstrated associations between prolonged preoperative wait time and poorer postoperative functional outcomes [23]. However, associations between preoperative wait time and short-term prognosis following intracranial tumor surgery remain largely unexplored—a critical knowledge gap given the potential for rapid neurological deterioration in this population.

Therefore, our objective was to investigate associations between preoperative wait time and postoperative thirty-day mortality following intracranial tumor craniotomy in adult patients. Utilizing data from the American College of Surgeons National Surgical Quality Improvement Program (ACS NSQIP) database, this retrospective cohort study evaluates the prognostic significance of preoperative wait time while adjusting for established risk factors. The findings may provide evidence-based guidance for optimizing surgical scheduling strategies and improving postoperative patient outcomes in neurosurgical practice.

## Materials and methods

### Study design and data source

This research is a retrospectively designed cohort study. We conducted a secondary analysis based on the ACS NSQIP database (2012–2015) originally made public by Zhang et al (Data source: https://doi.org/10.1371/journal.pone.0235273) [11]. The original study was freely released under a Creative Commons Attribution License, allowing unlimited use with proper attribution. Therefore, conducting a secondary analysis of this publicly accessible dataset does not contravene the original authors' copyright. The ACS NSQIP database comprises a representative sampling of hospitalized and ambulatory patients subjected to non-trauma surgery at around 400 academic and community hospitals in the United States. Collected information includes perioperative risk variables, comorbidities, surgical interventions executed utilizing Current Procedural Terminology (CPT) codes, and complications manifesting within the thirty-day postoperative interval following the index procedure.

### Ethics statement

The requirement of ethical approval was waived by the Ethics Committee of The First People's Hospital of Xiaoshan District for the studies involving humans because this study represents a secondary analysis utilizing a published public database with retrospective analytical nature. The studies were conducted in accordance with the local legislation and institutional requirements. The ethics committee also waived the requirement of written informed consent for participation because this study is based on a de-identified database, and the original personal information was anonymous.

### Study population

According to the inclusion criteria outlined in the original study, the data set was constructed using these specific CPT codes: 61510, 61521, 61520, 61518, 61526, 61545, 61546, 61512, 61519, and 61575 (refer to S1 Table for details), and comprised 18,642 adult patients who underwent intracranial tumor craniotomy between 2012 and 2015 [11]. Initially, patients with missing data on preoperative length of admission or survival status at 30 days postoperatively were excluded from the analysis. Subsequently, we excluded 88 patients with wait times exceeding 14 days (deemed outliers, defined as exceeding the mean wait time by ± 3 standard deviations). Then, we excluded 90 patients missing functional health status data and 166 patients missing ASA classification data. Finally, 18,298 patients were included for analysis (refer to Fig 1).

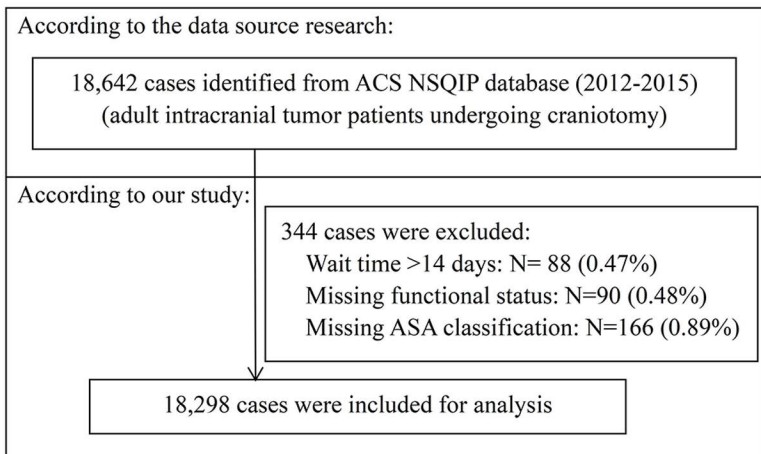

**Fig 1. Flowchart of study population selection.**

## Variables

Primary Exposure: Preoperative wait time was defined as the interval (in days) from hospital admission to surgical intervention. It was captured in the ACS NSQIP database as "days from hospital admission to operation". For analytical purposes, preoperative wait time was examined both as a continuous variable and as a categorical variable. For the latter approach, wait time was stratified into three groups: < 1 day (same-day surgery), 1–7 days, and >7 days, with <1 day serving as the reference group.

Primary Outcome: Postoperative thirty-day mortality was defined as all-cause death occurring within 30 days after the surgical procedure. This outcome was determined through the metric recorded as "days from operation to death" in the ACS NSQIP database. In the ACS NSQIP database, all patients underwent a thirty-day postoperative follow-up, and the vital status was continuously assessed until 30 days after surgery regardless of discharge status or length of hospital stay. The survival duration was determined as either the days survived after surgery for those who died within this period or 30 days for those who remained alive at the end of follow-up.

Covariates: The choice of covariates was based on clinical experience and prior research. We included the following covariates, categorized into preoperative and intraoperative domains: (1) Preoperative Factors: Demographic characteristics: sex (male/female), age ranges (years)(18–40, 41–60, 60–80, >80), race (white, black, asian, native, unknown), and smoking status; Clinical characteristics: BMI (calculated as weight/height squared (kg/m2)), ventilator dependent, functional health status (independent, partially dependent, totally dependent), steroid use for chronic condition, preoperative blood transfusion, and ASA classification (no disturb, mild disturb, severe disturb, life threat, moribund); Laboratory indicators: serum sodium (Na), blood urea nitrogen (BUN), white blood cell (WBC) counts, hematocrit (HCT), and international normalized ratio (INR); Comorbidities: diabetes, hypertension, severe chronic obstructive pulmonary disease (COPD), congestive heart failure (CHF), renal failure/dialysis, disseminated cancer, open wound infection, preoperative systemic infection, and bleeding disorders. (2) Intraoperative Factors: tumor type (uncertain type tumor, meningioma, cerebellopontine angle tumor, craniopharyngioma, pituitary macroadenoma), surgical site (supratentorial, infratentorial or posterior fossa, sellar region, others), operation time, emergency case, and wound classification (clean, clean-contaminated, contaminated, dirty/infected). Notably, the variables 'surgical site' and 'tumor type' were indirectly inferred from the CPT codes linked to each surgical procedure (S1 Table). Refer to the original research for further information [11].

## Statistical analysis

Our analytical approach involved several sequential steps to comprehensively assess the relationship between preoperative wait time and thirty-day mortality while accounting for potential confounders.

Missing data processing: After excluding patients with missing functional health status data and ASA classification data as described in the inclusion/exclusion criteria, all categorical variables in the final analysis sample had no missing values. The continuous variables with missing data included: BMI (N = 698, 3.81%), Na (N = 793, 4.33%), BUN (N = 1,516, 8.29%), WBC (N = 588, 3.21%), HCT (N = 434, 2.37%), INR (N = 2,755, 15.06%), and operation time (N = 2, 0.01%). The missing data were imputed using multivariate multiple imputation through a chained equation approach within the R MI procedure, based on 5 replications [24].

Descriptive statistics: Baseline characteristics were summarized using descriptive statistics. For normally distributed continuous variables, the mean and standard deviation (SD) were reported; for those with non-normal distributions, the median and interquartile range were provided. For categorical variables, the counts and their respective percentages were presented. Variations among the wait time categories were analyzed using One-way ANOVA for means, the Kruskal-Wallis H test for medians, and the chi-square test for percentages.

Univariate analysis: The association between all variables and thirty-day mortality was initially examined using unadjusted univariate logistic regression. Crude odds ratios (ORs) with 95% confidence intervals (CIs) were reported.

Selection of confounding variables: The confounding variables for further analysis were selected based on the following three aspects: clinical experience, literature reports, and statistical results. Statistically, covariates were included as potential confounders in the final models if they changed the effect estimates of wait time on thirty-day mortality by more than 10% or were significantly associated with thirty-day mortality [25]. The final selected confounding variables included: sex, age, tumor type, functional status, ventilator dependent, hypertension, diabetes, COPD, renal failure/dialysis, CHF, disseminated cancer, steroid usage, preoperative systemic infection, open wound infection, bleeding disorders, preoperative blood transfusion, emergency case, wound classification, ASA classification, operation time, Na, WBC, HCT, BUN, and INR.

Multivariate analysis: Due to the time-dependent nature of the outcome (the thirty-day mortality required evaluation of both event occurrence and time-to-event), Cox proportional hazards models were employed in multivariate regression analyses after verifying the proportional hazards assumption to evaluate the independent effect of wait time on the risk of thirty-day mortality. In accordance with the Strengthening the Reporting of Observational Studies in Epidemiology (STROBE) statement guidelines [26], three separate models were created: basic model with no adjustments (Crude model); minimally adjusted model (Model I), adjusted only for sex and age; fully adjusted model (Model II), adjusted for all confounding variables. The study included hazard ratios (HRs) as measures of effect and reported 95% confidence intervals (CIs).

Smooth curve fitting and Kaplan-Meier analysis: The liner or nonliner associations between wait time and thirty-day mortality were evaluated through smooth curve fitting using a restricted cubic spline [27]. The effects of wait time categories on thirty-day mortality were evaluated using Kaplan-Meier curves generated with the log-rank test.

Subgroup analyses: To examine whether the primary results were consistent across different subgroups of confounding factors, subgroup analyses using stratified Cox proportional hazards models were conducted. The model adjusted for all confounding variables except for the corresponding stratification variable. To evaluate whether these subgroup variables had an effect modification on the exposure-outcome association, interaction terms between wait time and subgroup variables were analyzed using likelihood ratio testing.

Sensitivity analyses: To evaluate the robustness of the main results, we conducted extensive sensitivity analyses. First, we performed a categorical multivariate regression analysis and trend analysis based on three wait time groups. Second, we constructed two new datasets by excluding all missing data and imputing missing data with the mean/median, named the complete dataset and the mean/median imputation dataset, respectively. Multivariate regression analyses were repeated

on the two new datasets. Third, we calculated E-values to assess the potential influence of unmeasured confounding on the relationship between preoperative wait time and thirty-day mortality. The E-values quantify the minimum strength of association that an unmeasured confounder must have with both the exposure and the outcome to completely account for the observed association [28]. Fourth, we employed another method—propensity score matching—to control for confounding factors and establish patient cohorts with comparable baseline characteristics based on wait time groups [29,30].

Patients were stratified into two wait time groups: < 1 day and ≥1 day. The variables used for matching included all previously identified confounding variables. The two groups were matched 1:1 without replacement based on propensity score using greedy matching with a 0.01 caliper width. Standardized differences less than 0.1 indicate small differences in matched variables between groups. The differences of thirty-day mortality between the matched two groups were compared using the McNemar test. Generalized estimating equations determined the percentage of absolute risk differences along with 95% CIs. In addition, three multivariate Cox regression analysis models were established after matching to assess the association between wait time and thirty-day mortality: an unadjusted crude model; Model I fully adjusted for all confounding variables; and Model II adjusted for propensity score. All results were reported under the guide of the STROBE statement [26,31].

The data analyses were conducted using two software tools: R (http://www.R-project.org; The R Foundation; version 4.2.0) and EmpowerStats (www.empowerstats.net, X&Y Solutions, Inc., Boston, MA). The statistical significance was set at a two-sided P-value < 0.05.

## Results

### Baseline characteristics of participants

Table 1 presents the baseline characteristics of 18,298 patients undergoing craniotomy, stratified by wait time into 3 groups: < 1 day (N = 10,914), 1–7 days (N = 6,880), and > 7 days (N = 504). Median wait times were 0, 3, and 9 days for the 3 groups, respectively. Most patients were females, aged 41–80 years, of white race, non-smokers, with supratentorial site and uncertain type tumors, functionally independent, not ventilator dependent, non-emergency cases, without steroid use, blood transfusion, and major comorbidities. The overall postoperative thirty-day mortality rate was 2.42% (442/18,298). Trends were observed across groups for all continuous variables: patients with longer wait times showed graded decreases in preoperative BMI, Na, HCT, and operation time (all P < 0.001). In contrast, preoperative BUN, WBC, and INR increased with longer wait times (all P < 0.001). Additionally, the groups with longer wait times had larger proportions of subpopulations including males, elderly patients (age > 60 years), non-whites, current smokers, dependent functional status, ventilator dependence, higher wound contamination class and ASA scores, and major comorbidities like COPD, diabetes, hypertension, CHF, disseminated cancer, renal dysfunction, preoperative infection, open wound infection, bleeding disorders, and blood transfusion (all P < 0.005). The distribution of surgical site, tumor type, steroid use, and emergency case also differed significantly across wait time groups (all P < 0.001).

### Univariate regression analyses

S2 Table depicts the associated covariates with thirty-day mortality following craniotomy identified through univariate analyses. Increased preoperative wait time conferred a markedly higher mortality risk with a 17.6% rise per day (OR=1.176, 95%CI: 1.142–1.210), representing one of the most substantial associations observed. Laboratory indicators including increased BUN, WBC, and INR, as well as decreased Na and HCT, were each associated with elevated risk of mortality (all P < 0.001). Longer operation time showed a protective trend (P < 0.001). Regarding demographics, females conferred a more favorable outcome, whereas mortality rates increased with advancing age (both P < 0.001). The comorbidities such as hypertension, diabetes, COPD, CHF, renal dysfunction, disseminated cancer, preoperative infection, bleeding disorders, and open wound infection were strongly associated with elevated risk of mortality (all P < 0.001), along with indicators of frailty like worse functional status and higher ASA grade (both P < 0.001). Other factors including ventilator

**Table 1. Baseline characteristics of study population (N = 18,298).**

| Wait Time (days) | <1 | 1-7 | >7 | P-value |
|---|---|---|---|---|
| N (cases) | 10914 | 6880 | 504 | |
| **Primary Exposure** | | | | |
| Wait time (day, Median (Q1-Q3)) | 0.00 (0.00-0.00) | 3.00 (1.00-4.00) | 9.00 (8.00-11.00) | <0.001 |
| **Demographic characteristics** | | | | |
| Sex, N (%) | | | | <0.001 |
| male | 4985 (45.68) | 3411 (49.58) | 257 (50.99) | |
| female | 5929 (54.32) | 3469 (50.42) | 247 (49.01) | |
| Age ranges (years), N (%) | | | | <0.001 |
| 18-40 | 2025 (18.55) | 905 (13.15) | 54 (10.71) | |
| 41-60 | 4699 (43.05) | 2757 (40.07) | 181 (35.91) | |
| 61-80 | 3917 (35.89) | 2924 (42.50) | 243 (48.21) | |
| >80 | 273 (2.50) | 294 (4.27) | 26 (5.16) | |
| Race, N (%) | | | | <0.001 |
| White | 8160 (74.77) | 4651 (67.60) | 254 (50.40) | |
| Black | 636 (5.83) | 517 (7.51) | 64 (12.70) | |
| Asian | 312 (2.86) | 197 (2.86) | 17 (3.37) | |
| Native | 78 (0.71) | 28 (0.41) | 3 (0.60) | |
| Unknown | 1728 (15.83) | 1487 (21.61) | 166 (32.94) | |
| Smoking status, N (%) | | | | <0.001 |
| No | 9078 (83.18) | 5309 (77.17) | 365 (72.42) | |
| Yes | 1836 (16.82) | 1571 (22.83) | 139 (27.58) | |
| **Preoperative Laboratory Indicators** | | | | |
| Na (mmol/L, Mean ± SD) | 138.91 ± 3.01 | 138.30 ± 3.25 | 137.12 ± 3.61 | <0.001 |
| BUN (mg/dL, Mean ± SD) | 16.83 ± 7.00 | 18.00 ± 8.75 | 20.92 ± 11.79 | <0.001 |
| WBC (10^9/L, Mean ± SD) | 8.71 ± 4.00 | 10.60 ± 4.69 | 11.57 ± 4.86 | <0.001 |
| HCT (%, Mean ± SD) | 41.06 ± 4.34 | 39.33 ± 5.00 | 38.49 ± 6.07 | <0.001 |
| INR (ratio, Mean ± SD) | 1.01 ± 0.18 | 1.04 ± 0.22 | 1.05 ± 0.21 | <0.001 |
| **Preoperative Clinical Characteristics** | | | | |
| BMI (kg/m2, Mean ± SD) | 28.97 ± 6.71 | 28.45 ± 6.43 | 28.41 ± 6.58 | <0.001 |
| Functional health status, N (%) | | | | <0.001 |
| Independent | 10619 (97.30) | 6487 (94.29) | 453 (89.88) | |
| Partially Dependent | 261 (2.39) | 342 (4.97) | 45 (8.93) | |
| Totally Dependent | 34 (0.31) | 51 (0.74) | 6 (1.19) | |
| Ventilator dependent, N (%) | | | | <0.001 |
| No | 10861 (99.51) | 6743 (98.01) | 494 (98.02) | |
| Yes | 53 (0.49) | 137 (1.99) | 10 (1.98) | |
| Steroid use for chronic condition, N (%) | | | | <0.001 |
| No | 8997 (82.44) | 6149 (89.38) | 425 (84.33) | |
| Yes | 1917 (17.56) | 731 (10.62) | 79 (15.67) | |
| Preoperative blood transfusion, N (%) | | | | <0.001 |
| No | 10905 (99.92) | 6843 (99.46) | 493 (97.82) | |
| Yes | 9 (0.08) | 37 (0.54) | 11 (2.18) | |
| ASA classification, N (%) | | | | <0.001 |
| No Disturb | 148 (1.36) | 94 (1.37) | 4 (0.79) | |
| Mild Disturb | 3490 (31.98) | 1229 (17.86) | 37 (7.34) | |
| Severe Disturb | 6330 (58.00) | 4178 (60.73) | 335 (66.47) | |

*(Continued)*

**Table 1.** (Continued)

| Wait Time (days) | <1 | 1-7 | >7 | P-value |
|---|---|---|---|---|
| Life Threat | 924 (8.47) | 1365 (19.84) | 127 (25.20) | |
| Moribund | 22 (0.20) | 14 (0.20) | 1 (0.20) | |
| **Preoperative Comorbidities** | | | | |
| Severe COPD, N (%) | | | | <0.001 |
| No | 10513 (96.33) | 6521 (94.78) | 456 (90.48) | |
| Yes | 401 (3.67) | 359 (5.22) | 48 (9.52) | |
| Diabetes, N (%) | | | | <0.001 |
| No | 9797 (89.77) | 5977 (86.88) | 403 (79.96) | |
| Yes(Insulin) | 399 (3.66) | 339 (4.93) | 40 (7.94) | |
| Yes(Oral) | 718 (6.58) | 564 (8.20) | 61 (12.10) | |
| Hypertension, N (%) | | | | <0.001 |
| No | 6987 (64.02) | 4098 (59.56) | 240 (47.62) | |
| Yes | 3927 (35.98) | 2782 (40.44) | 264 (52.38) | |
| Congestive heart failure, N (%) | | | | 0.002 |
| No | 10891 (99.79) | 6854 (99.62) | 499 (99.01) | |
| Yes | 23 (0.21) | 26 (0.38) | 5 (0.99) | |
| Renal failure/Dialysis, N (%) | | | | <0.001 |
| No | 10895 (99.83) | 6842 (99.45) | 500 (99.21) | |
| Yes | 19 (0.17) | 38 (0.55) | 4 (0.79) | |
| Disseminated cancer, N (%) | | | | <0.001 |
| No | 9185 (84.16) | 4814 (69.97) | 348 (69.05) | |
| Yes | 1729 (15.84) | 2066 (30.03) | 156 (30.95) | |
| Open wound infection, N (%) | | | | <0.001 |
| No | 10861 (99.51) | 6800 (98.84) | 487 (96.63) | |
| Yes | 53 (0.49) | 80 (1.16) | 17 (3.37) | |
| Preoperative systemic infection, N (%) | | | | <0.001 |
| No | 10824 (99.18) | 6371 (92.60) | 458 (90.87) | |
| SIRS | 85 (0.78) | 479 (6.96) | 38 (7.54) | |
| Sepsis/Septic Shock | 5 (0.05) | 30 (0.44) | 8 (1.59) | |
| Bleeding disorders, N (%) | | | | <0.001 |
| No | 10766 (98.64) | 6682 (97.12) | 488 (96.83) | |
| Yes | 148 (1.36) | 198 (2.88) | 16 (3.17) | |
| **Intraoperative Characteristics** | | | | |
| Operation time (minutes, Median (Q1-Q3)) | 190.00 (126.00-282.00) | 170.00 (115.00-245.00) | 159.50 (103.75-241.00) | <0.001 |
| Surgical site, N (%) | | | | <0.001 |
| Supratentorial | 8450 (77.42) | 5447 (79.17) | 398 (78.97) | |
| Infratentorial or posterior fossa | 2240 (20.52) | 1346 (19.56) | 95 (18.85) | |
| Sellar region | 169 (1.55) | 64 (0.93) | 11 (2.18) | |
| Others | 55 (0.50) | 23 (0.33) | 0 (0.00) | |
| Tumor type, N (%) | | | | <0.001 |
| Uncertain type tumor | 6934 (63.53) | 5465 (79.43) | 389 (77.18) | |
| Meningioma | 2999 (27.48) | 1123 (16.32) | 86 (17.06) | |
| Cerebellopontine angle tumor | 812 (7.44) | 228 (3.31) | 18 (3.57) | |
| Craniopharyngioma | 62 (0.57) | 29 (0.42) | 7 (1.39) | |
| Pituitary macroadenoma | 107 (0.98) | 35 (0.51) | 4 (0.79) | |
| Emergency case, N (%) | | | | <0.001 |

*(Continued)*

**Table 1.** (Continued)

| Wait Time (days) | <1 | 1-7 | >7 | P-value |
|---|---|---|---|---|
| No | 10645 (97.54) | 6015 (87.43) | 467 (92.66) | |
| Yes | 269 (2.46) | 865 (12.57) | 37 (7.34) | |
| Wound classification, N (%) | | | | <0.001 |
| Clean | 10628 (97.38) | 6673 (96.99) | 487 (96.63) | |
| Clean-Contaminated | 154 (1.41) | 59 (0.86) | 5 (0.99) | |
| Contaminated | 108 (0.99) | 110 (1.60) | 8 (1.59) | |
| Dirty/Infected | 24 (0.22) | 38 (0.55) | 4 (0.79) | |
| **Primary Outcome** | | | | |
| Thirty-day mortality, N (%) | | | | <0.001 |
| No | 10758 (98.57) | 6629 (96.35) | 469 (93.06) | |
| Yes | 156 (1.43) | 251 (3.65) | 35 (6.94) | |

BMI: Body-mass index; Na: Serum sodium; BUN: blood urea nitrogen; WBC: White blood cells; HCT: hematocrit; INR: International normalized ratio; COPD: chronic obstructive pulmonary disease; SD: standard deviation.

dependence, steroid usage, preoperative blood transfusion, emergency surgery, and dirty/infected wounds were also associated with elevated risk of mortality (all P < 0.001). Of note, different tumor types significantly affect mortality, whereas smoking, BMI, race, and surgical site did not have a significant influence on mortality (all P > 0.05).

## Multivariate regression analyses

Table 2 depicts the independent exposure-outcome association assessed by multivariate regression analyses. The Crude model (no adjustment) showed a 17.1% rise in the risk of thirty-day mortality for each additional day of waiting (HR = 1.171, 95%CI: 1.139–1.204). Model I (minimal adjustments) indicated a 14.8% increase in mortality risk for each day of waiting (HR = 1.148, 95%CI: 1.116–1.182). Finally, Model II (full adjustments) revealed an attenuated yet notable 7.5% higher risk of thirty-day mortality for each prolonged day of waiting (HR = 1.075, 95%CI: 1.040–1.110). The 95%CI indicates a dependable association between wait time and thirty-day mortality.

**Table 2.** The multivariate analyses of the association between wait time and thirty-day mortality.

| Exposure | Crude model | Model I | Model II |
|---|---|---|---|
| | HR (95% CI) P-value | HR (95% CI) P-value | HR (95% CI) P-value |
| Wait time | 1.171 (1.139, 1.204) < 0.00001 | 1.148 (1.116, 1.182) < 0.00001 | 1.075 (1.040-1.110) 0.00001 |
| Wait time group | | | |
| <1 | Ref | Ref | Ref |
| 1-7 | 2.580 (2.113, 3.151) < 0.00001 | 2.289 (1.873, 2.799) < 0.00001 | 1.519 (1.223, 1.886) 0.00015 |
| >7 | 5.021 (3.480, 7.244) < 0.00001 | 4.253 (2.945, 6.142) < 0.00001 | 2.072 (1.406, 3.052) 0.00023 |
| P for trend | <0.001 | <0.001 | <0.001 |

HR, hazard ratio; 95% CI, 95% confidence interval; Ref, reference.

Crude model (non-adjusted model): adjusted for none.

Model I (minimally adjusted model): adjusted for sex and age.

Model II (fully adjusted model): adjusted for sex, age, tumor type, functional status, ventilator dependent, COPD, diabetes, hypertension, CHF, renal failure/dialysis, disseminated cancer, steroid use, preoperative systemic infection, open wound infection, bleeding disorders, preoperative blood transfusion, emergency case, wound classification, ASA classification, Na, BUN, WBC, HCT, INR and operation time.

## Smooth curve fitting and Kaplan-Meier analysis

Fig 2 displays a linear association between preoperative wait time and postoperative thirty-day mortality. Consistent with the results of the multivariate analysis, the thirty-day mortality risk gradually rises with increased wait times.

Fig 3 displays the results of Kaplan–Meier analysis, demonstrating a significantly higher overall cumulative hazard of thirty-day mortality in patients with longer wait times (1–7 days and > 7 days) compared to those waited less than one day (P < 0.001).

## Subgroup analyses and interaction testing

Table 3 depicts the impact of some stratified confounding factors on the association between preoperative wait time and postoperative thirty-day mortality. The prognostic influence of longer wait time was mainly consistent across most subgroups of confounding factors, including sex, age, tumor type, functional health status, diabetes, hypertension, metastatic cancer, steroid usage, ASA classification, and emergency case. The adjusted hazard ratios ranged from 0.776 to 1.242 across these subgroups, with most 95%CIs excluding the null value of 1. However, the prognostic impact of wait time showed a different trend in the subgroups of confounding factors such as ventilator dependent, severe COPD, bleeding disorders, and preoperative systemic infection. Nevertheless, the differences in the above subgroups, apart from ventilator dependent, were not statistically significant (P for interaction > 0.05). A significant interaction was observed in the ventilator dependence subgroup (P for interaction = 0.0360). For non-ventilator-dependent patients, each day prolonged in wait time conferred an 8.3% higher risk of thirty-day mortality (HR = 1.083, 95%CI: 1.049–1.119). In contrast, longer wait times did not confer a higher mortality risk in ventilator-dependent patients (HR = 0.773, 95%CI: 0.561–1.074). No other significant interaction effects were observed (all P > 0.05).

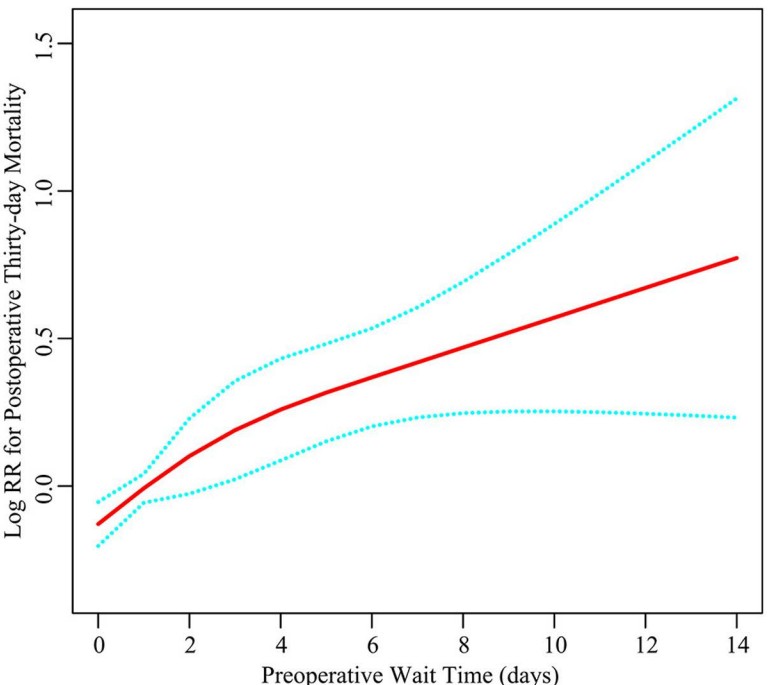

**Fig 2. The relationship between preoperative wait time and postoperative thirty-day mortality.** Red line: Log RR for postoperative thirty-day mortality; Blue line: 95%CI. The model adjusted for all confounding variables.

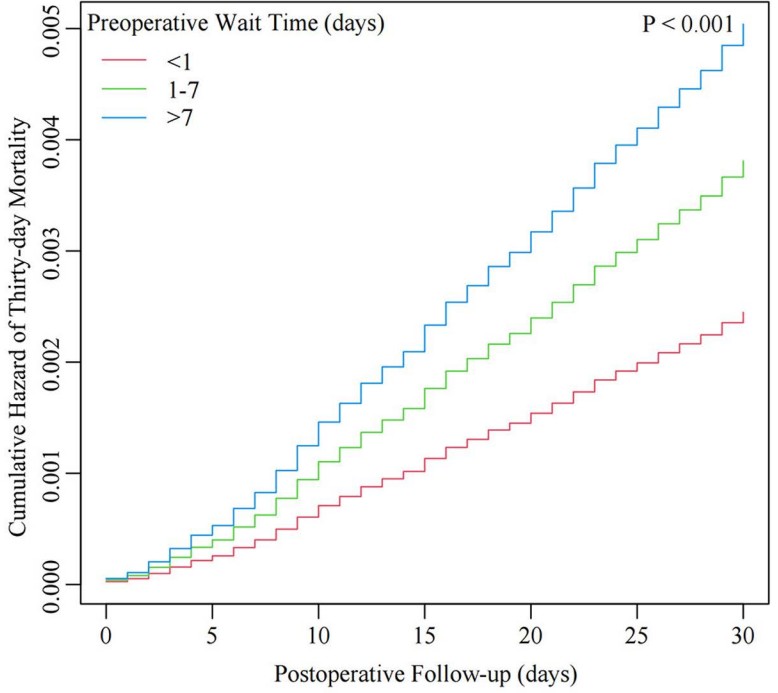

**Fig 3. The cumulative hazard curve of thirty-day mortality stratified by preoperative wait time.** Red line: <1 day group; Green line: 1-7 days group; Blue line: >7 days group. The model adjusted for all confounding variables.

## Propensity score matching and multivariate analyses after matching

Table 4 presents the baseline characteristics of patients before and after propensity score matching in the <1 day and ≥ 1 day wait time groups. Before matching, the standardized differences of most covariates between the two groups were greater than 0.1, indicating moderate or large differences in baseline characteristics. In contrast, the standardized differences of most covariates were less than 0.1 after matching, showing balance in baseline characteristics between the two groups. There were 5537 patients 1:1 matched between two groups, respectively. As depicted in Table 5, of the 5537 matched patients waiting <1 day for surgery, 122 patients (2.20%) died within thirty days after craniotomy, whereas 163 patients (2.94%) of 5537 died within thirty days in the ≥ 1 day group. The absolute risk difference in thirty-day mortality between the two groups was 0.74% (95% CI: 0.15–1.33, P < 0.0001).

Table 6 presents the subsequent multivariate analyses conducted in the propensity score-matched patient cohorts. The unadjusted Crude model showed a 34.1% higher thirty-day mortality risk of the ≥ 1 day group compared to the <1 day group (HR = 1.341, 95%CI: 1.060–1.695). Model I (fully adjusted for all covariates) indicated a 36.5% rise in the risk of thirty-day mortality for the ≥ 1 day group compared to the <1 day group (HR = 1.365, 95%CI: 1.075–1.734). Finally, Model II (adjusted for propensity score) indicated the ≥ 1 day group had a 31.3% higher risk of thirty-day mortality compared to the <1 day group (HR = 1.313, 95%CI: 1.039–1.661).

## Sensitivity analyses

To rigorously evaluate the robustness of our results, we conducted extensive sensitivity analyses. We first performed a categorical multivariate regression analysis (Table 2). In the model with full adjustments, the thirty-day mortality risk compared to the reference < 1 day group was 51.9% higher in the 1–7 days group (HR = 1.519, 95%CI: 1.223–1.886) and

**Table 3. The results of subgroup analyses and interaction testing.**

| Characteristic | N | HR (95% CI) | P for interaction |
|---|---|---|---|
| Sex | | | 0.8672 |
| male | 8653 | 1.070 (1.025, 1.117) | |
| female | 9645 | 1.086 (1.032, 1.144) | |
| Age ranges | | | 0.6699 |
| 18-60 | 10621 | 1.078 (1.018, 1.141) | |
| >60 | 7677 | 1.071 (1.030, 1.115) | |
| Functional health status | | | 0.3332 |
| Independent | 17559 | 1.072 (1.034, 1.111) | |
| Partially/Totally Dependent | 739 | 1.128 (1.035, 1.229) | |
| Ventilator dependent | | | 0.0360 |
| No | 18098 | 1.083 (1.049, 1.119) | |
| Yes | 200 | 0.776 (0.561, 1.074) | |
| Steroid use for chronic condition | | | 0.7350 |
| No | 15571 | 1.076 (1.035, 1.120) | |
| Yes | 2727 | 1.060 (0.996, 1.130) | |
| ASA classification | | | 0.1145 |
| No/Mild Disturb | 5002 | 1.242 (1.092, 1.411) | |
| Severe Disturb | 10843 | 1.069 (1.023, 1.117) | |
| Life Threat/Moribund | 2453 | 1.066 (1.011, 1.125) | |
| Severe COPD | | | 0.1319 |
| No | 17490 | 1.085 (1.049, 1.123) | |
| Yes | 808 | 0.988 (0.879, 1.110) | |
| Diabetes | | | 0.5750 |
| No | 16177 | 1.075 (1.036, 1.116) | |
| Yes (Insulin/Oral) | 2122 | 1.082 (1.005, 1.166) | |
| Hypertension | | | 0.6244 |
| No | 11325 | 1.073 (1.018, 1.132) | |
| Yes | 6973 | 1.073 (1.029, 1.119) | |
| Disseminated cancer | | | 0.6894 |
| No | 14347 | 1.090 (1.044, 1.138) | |
| Yes | 3951 | 1.070 (1.019, 1.127) | |
| Bleeding disorders | | | 0.4177 |
| No | 17936 | 1.075 (1.040, 1.112) | |
| Yes | 362 | 0.939 (0.826, 1.067) | |
| Preoperative systemic infection | | | 0.3054 |
| No | 17653 | 1.085 (1.049, 1.123) | |
| SIRS/Sepsis/Septic Shock | 645 | 0.988 (0.871, 1.120) | |
| Tumor type | | | 0.2912 |
| Uncertain type tumor | 12788 | 1.061 (1.022, 1.100) | |
| Meningioma | 4208 | 1.048 (0.939, 1.170) | |
| Cerebellopontine angle tumor/Craniopharyngioma/Pituitary macroadenoma | 1302 | 1.207 (1.057, 1.378) | |
| Emergency case | | | 0.4036 |
| No | 17127 | 1.082 (1.045, 1.120) | |
| Yes | 1171 | 1.022 (0.916, 1.141) | |

HR, hazard ratio; 95% CI, 95% confidence interval

Note: The model adjusted for all confounding variables except the corresponding stratification variable.

**Table 4. The results of propensity score matching based on wait time groups.**

| | Before Matching, No. (%) | | | After Matching, No. (%) | | |
|---|---|---|---|---|---|---|
| | <1 day (N = 10914) | ≥1day (N = 7384) | Standardized Difference | <1 day (N = 5537) | ≥1day (N = 5537) | Standardized Difference |
| Wait time to surgery (days, Mean ± SD) | 0.00 ± 0.00 | 3.45 ± 2.47 | | 0.00 ± 0.00 | 3.40 ± 2.46 | 1.9564 |
| **Demographic characteristics** | | | | | | |
| Sex, N (%) | | | 0.0801 | | | 0.0325 |
| male | 4985 (45.7) | 3668 (49.7) | | 2760 (49.8) | 2670 (48.2) | |
| female | 5929 (54.3) | 3716 (50.3) | | 2777 (50.2) | 2867 (51.8) | |
| Age ranges, N (%) | | | | | | |
| 18-40 | 2025 (18.6) | 959 (13.0) | 0.0916 | 684 (12.4) | 830 (15) | 0.0768 |
| 41-60 | 4699 (43.0) | 2938 (39.8) | 0.0666 | 2180 (39.4) | 2300 (41.5) | 0.0442 |
| 61-80 | 3917 (35.9) | 3167 (42.9) | 0.1554 | 2454 (44.3) | 2212 (39.9) | 0.0886 |
| >80 | 273 (2.5) | 320 (4.3) | 0.0714 | 219 (4) | 195 (3.5) | 0.0229 |
| **Preoperative Laboratory Indicators** | | | | | | |
| Na (Mean ± SD) | 138.91 ± 3.01 | 138.22 ± 3.29 | 0.2194 | 138.30 ± 3.24 | 138.44 ± 3.16 | 0.0446 |
| BUN (Mean ± SD) | 16.83 ± 7.00 | 18.20 ± 9.01 | 0.1697 | 17.97 ± 8.15 | 17.61 ± 8.19 | 0.0442 |
| WBC (Mean ± SD) | 8.71 ± 4.00 | 10.66 ± 4.70 | 0.4479 | 9.55 ± 4.65 | 10.19 ± 4.10 | 0.1473 |
| HCT (Mean ± SD) | 41.06 ± 4.34 | 39.27 ± 5.08 | 0.3793 | 39.95 ± 4.59 | 39.98 ± 4.75 | 0.007 |
| INR (Mean ± SD) | 1.01 ± 0.18 | 1.04 ± 0.22 | 0.1588 | 1.02 ± 0.24 | 1.04 ± 0.18 | 0.0646 |
| **Preoperative Clinical Characteristics** | | | | | | |
| Functional health status, N (%) | | | | | | |
| Independent | 10619 (97.3) | 6940 (94.0) | 0.0683 | 5287 (95.5) | 5266 (95.1) | 0.0179 |
| Partially Dependent | 261 (2.4) | 387 (5.2) | 0.1107 | 217 (3.9) | 245 (4.4) | 0.0253 |
| Totally Dependent | 34 (0.3) | 57 (0.8) | 0.0424 | 33 (0.6) | 26 (0.5) | 0.0174 |
| Ventilator dependent, N (%) | | | 0.1364 | | | 0.0543 |
| No | 10861 (99.5) | 7237 (98.0) | | 5486 (99.1) | 5453 (98.5) | |
| Yes | 53 (0.5) | 147 (2.0) | | 51 (0.9) | 84 (1.5) | |
| Steroid use for chronic condition, N (%) | | | 0.1894 | | | 0.0054 |
| No | 8997 (82.4) | 6574 (89.0) | | 4843 (87.5) | 4833 (87.3) | |
| Yes | 1917 (17.6) | 810 (11.0) | | 694 (12.5) | 704 (12.7) | |
| Preoperative blood transfusion, N (%) | | | 0.0941 | | | 0.0539 |
| No | 10905 (99.9) | 7336 (99.4) | | 5529 (99.9) | 5513 (99.6) | |
| Yes | 9 (0.1) | 48 (0.6) | | 8 (0.1) | 24 (0.4) | |
| ASA classification, N (%) | | | | | | |
| No Disturb | 148 (1.3) | 98 (1.3) | 0.0094 | 20 (0.4) | 95 (1.7) | 0.1339 |
| Mild Disturb | 3490 (32.0) | 1266 (17.2) | 0.2676 | 1038 (18.7) | 1156 (20.9) | 0.0535 |
| Severe Disturb | 6330 (58.0) | 4513 (61.1) | 0.1131 | 3695 (66.7) | 3455 (62.4) | 0.0907 |
| Life Threat | 924 (8.5) | 1492 (20.2) | 0.3703 | 762 (13.8) | 826 (14.9) | 0.033 |
| Moribund | 22 (0.2) | 15 (0.2) | 0.0000 | 22 (0.4) | 5 (0.1) | 0.0623 |
| **Preoperative Comorbidities** | | | | | | |
| Severe COPD, N (%) | | | 0.0879 | | | 0.0008 |
| No | 10513 (96.3) | 6977 (94.5) | | 5260 (95) | 5259 (95) | |
| Yes | 401 (3.7) | 407 (5.5) | | 277 (5) | 278 (5) | |
| Diabetes, N (%) | | | | | | |
| No | 9797 (89.8) | 6380 (86.4) | 0.0915 | 4777 (86.3) | 4845 (87.5) | 0.0364 |
| Yes (Insulin) | 399 (3.7) | 379 (5.1) | 0.0976 | 276 (5) | 265 (4.8) | 0.0092 |
| Yes (Oral) | 718 (6.6) | 625 (8.5) | 0.0983 | 484 (8.7) | 427 (7.7) | 0.0375 |

*(Continued)*

**Table 4.** (Continued)

| | Before Matching, No. (%) | | | After Matching, No. (%) | | |
|---|---|---|---|---|---|---|
| | <1 day (N = 10914) | ≥1day (N = 7384) | Standardized Difference | <1 day (N = 5537) | ≥1day (N = 5537) | Standardized Difference |
| Hypertension, N (%) | | | 0.1084 | | | 0.0582 |
| No | 6987 (64.0) | 4338 (58.8) | | 3226 (58.3) | 3384 (61.1) | |
| Yes | 3927 (36.0) | 3046 (41.3) | | 2311 (41.7) | 2153 (38.9) | |
| Congestive heart failure, N (%) | | | 0.0373 | | | 0.0094 |
| No | 10891 (99.8) | 7353 (99.6) | | 5517 (99.6) | 5520 (99.7) | |
| Yes | 23 (0.2) | 31 (0.4) | | 20 (0.4) | 17 (0.3) | |
| Renal failure/Dialysis, N (%) | | | 0.0649 | | | 0.0065 |
| No | 10895 (99.8) | 7342 (99.4) | | 5519 (99.7) | 5521 (99.7) | |
| Yes | 19 (0.2) | 42 (0.6) | | 18 (0.3) | 16 (0.3) | |
| Disseminated cancer, N (%) | | | 0.3438 | | | 0.0554 |
| No | 9185 (84.2) | 5162 (69.9) | | 4114 (74.3) | 4246 (76.7) | |
| Yes | 1729 (15.8) | 2222 (30.1) | | 1423 (25.7) | 1291 (23.3) | |
| Open wound infection, N (%) | | | 0.0878 | | | 0.0059 |
| No | 10861 (99.5) | 7287 (98.7) | | 5491 (99.2) | 5488 (99.1) | |
| Yes | 53 (0.5) | 97 (1.3) | | 46 (0.8) | 49 (0.9) | |
| Preoperative systemic infection, N (%) | | | 0.3395 | | | 0.1563 |
| No | 10824 (99.2) | 6829 (92.5) | | 5452 (98.5) | 5309 (95.9) | |
| SIRS/Sepsis/Septic Shock | 90 (0.8) | 555 (7.5) | | 85 (1.5) | 228 (4.1) | |
| Bleeding disorders, N (%) | | | 0.1070 | | | 0.0073 |
| No | 10766 (98.6) | 7170 (97.1) | | 5415 (97.8) | 5409 (97.7) | |
| Yes | 148 (1.4) | 214 (2.9) | | 122 (2.2) | 128 (2.3) | |
| **Intraoperative Characteristics** | | | | | | |
| Operation time (Mean ± SD) | 221.31 ± 135.70 | 196.38 ± 123.96 | 0.1918 | 200.54 ± 124.58 | 205.24 ± 129.15 | 0.0371 |
| Tumor type, N (%) | | | | | | |
| Uncertain type tumor | 6934 (63.5) | 5854 (79.3) | 0.3586 | 4382 (79.1) | 4177 (75.4) | 0.0885 |
| Meningioma | 2999 (27.5) | 1209 (16.4) | 0.2697 | 979 (17.7) | 1058 (19.1) | 0.0368 |
| Cerebellopontine angle tumor | 812 (7.4) | 246 (3.3) | 0.1822 | 156 (2.8) | 231 (4.2) | 0.0738 |
| Craniopharyngioma | 62 (0.6) | 36 (0.5) | 0.0110 | 8 (0.1) | 33 (0.6) | 0.0744 |
| Pituitary macroadenoma | 107 (1.0) | 39 (0.5) | 0.0519 | 12 (0.2) | 38 (0.7) | 0.0701 |
| Emergency case, N (%) | | | 0.3806 | | | 0.105 |
| No | 10645 (97.5) | 6482 (87.8) | | 5273 (95.2) | 5135 (92.7) | |
| Yes | 269 (2.5) | 902 (12.2) | | 264 (4.8) | 402 (7.3) | |
| Wound classification, N (%) | | | | | | |
| Clean | 10628 (97.4) | 7160 (97.9) | 0.0313 | 5374 (97.1) | 5389 (97.3) | 0.0164 |
| Clean-Contaminated | 154 (1.4) | 64 (0.9) | 0.0860 | 73 (1.3) | 51 (0.9) | 0.0378 |
| Contaminated | 108 (1.0) | 118 (1.6) | 0.0745 | 71 (1.3) | 75 (1.4) | 0.0063 |
| Dirty/Infected | 24 (0.2) | 42 (0.6) | 0.0717 | 19 (0.3) | 22 (0.4) | 0.0089 |

Note: Standardized differences: less than 0.1 indicate small differences in covariates between groups; between 0.1 and 0.2 indicate moderate differences; greater than 0.2 indicate large differences.

**Table 5. Thirty-day mortality of two wait time groups after propensity score matching.**

| Outcome | No. (%) of Patients | | | |
|---|---|---|---|---|
| | <1 day (N=5537) | ≥1day (N=5537) | Absolute Risk Difference, % (95% CI) | P Value |
| Thirty-day mortality, N (%) | 122 (2.20) | 163 (2.94) | 0.74 (0.15, 1.33) | <0.0001 |

95% CI, 95% confidence interval.

Note: P values were calculated using the McNemar test.

**Table 6. The result of multivariate analyses after propensity score matching.**

| Exposure | Crude model | Model I | Model II |
|---|---|---|---|
| | HR (95% CI) P-value | HR (95% CI) P-value | HR (95% CI) P-value |
| Wait time group | | | |
| < 1 day | Ref | Ref | Ref |
| ≥1 day | 1.341 (1.060, 1.695) 0.01434 | 1.365 (1.075, 1.734) 0.01066 | 1.313 (1.039, 1.661) 0.02281 |

HR, hazard ratio; 95% CI, 95% confidence interval; Ref, reference.

Crude model: adjusted for none. Model I: adjusted for all confounding variables. Model II: adjusted for propensity score.

107.2% higher in the >7 days group (HR=2.072, 95%CI: 1.406–3.052). Trend analysis revealed statistically significant differences in thirty-day mortality across the wait time groups (P<0.001).

In addition, we reanalyzed data from 14,781 complete cases after excluding 3517 cases with missing data (S3 Table). Among all complete cases, the thirty-day mortality rate was 2.65% (392/14,781). The multivariate analysis indicated a 7.2% rise in the risk of thirty-day mortality for each additional day of waiting (HR=1.072, 95%CI: 1.036–1.110). In categorical analysis, the thirty-day mortality risk of the 1–7 days group and > 7 days group was 45.9% and 101.4% higher respectively compared to the <1 day group.

Furthermore, we used mean/median to impute missing data of covariates (S3 Table). The multivariate analysis of mean/median imputation datasets showed a 7.5% higher thirty-day mortality risk for each additional day of wait time (HR=1.075, 95%CI: 1.040–1.110), and the categorical analysis showed 51.3% and 106.5% higher mortality risk for the 1–7 days group and > 7 days group, respectively.

Moreover, we calculated an E-value to assess the sensitivity to unmeasured confounders. The E-value of 1.36 demonstrates the robustness of our primary findings. The observed association between wait time and thirty-day mortality would not be nullified by unmeasured confounding unless there exists evidence of confounding factors that have an association strength (risk ratio) of at least 1.36 with both the exposure and the outcome.

All these results from the sensitivity analyses aligned with our previous main analyses, further corroborating the robustness of the main results.

## Discussion

Analyzing data of 18,298 patients extracted from the ACS NSQIP database, this large retrospective cohort study investigated associations between preoperative wait time and postoperative thirty-day mortality following intracranial tumor craniotomy in adult patients. Our results indicate that prolonged wait time was independently and linearly associated with elevated postoperative thirty-day mortality. In addition, the linear wait time-mortality association was observed solely in non-ventilator-dependent patients, showing an 8.3% increase in thirty-day mortality risk for each additional day of waiting

(HR = 1.083, 95%CI: 1.049–1.119). Furthermore, patients who waited ≥ 1 day had a 0.74% higher absolute risk and a 31.3% higher relative risk of thirty-day mortality compared to those who waited < 1 day. Sensitivity analyses corroborated the robustness of these findings.

Previous research has found a relationship between preoperative wait time and postoperative outcomes across various diseases, suggesting it could be a potential risk factor for patient prognosis in conditions such as head and neck, breast, and colon cancer [20,21], lung cancer [22], cervical spondylotic myelopathy [23], hip fracture [29], lumbar disc herniation [32], cervical cancer [33], benign gynecologic disease [34], and urinary tract urothelial carcinoma [35]. Together, these discoveries underscore the significance of wait time as a predictive indicator of patient prognosis across different clinical contexts. However, this connection was not observed in kidney cancer [36], esophageal cancer [37], and gastric cancer [38]. To date, associations between wait time and short-term prognosis following intracranial tumor surgery have not been investigated, thus our study sought to explore. In addition, prior research has largely defined preoperative wait time as the duration from diagnosis to surgical intervention, encompassing both pre-hospital and in-hospital phases. Pre-hospital wait time is influenced by multiple factors and is challenging to intervene, whereas in-hospital wait time is more readily manageable by clinicians [39]. In contrast, our study defines preoperative wait time as the duration from hospital admission to surgical intervention, focusing on the association between in-hospital wait time and short-term postoperative outcomes.

Consistent with findings across various clinical contexts, our study extends the predictive value of preoperative wait time to neurosurgical patients with intracranial tumor. To our knowledge, this is the first study in an American population to demonstrate that preoperative wait time can independently predict short-term prognosis following intracranial tumor craniotomy in non-ventilator-dependent adult patients. Our results show that the thirty-day mortality risk is significantly elevated even when the preoperative wait time exceeds only one day. This suggests that clinicians should strive to minimize preoperative wait time after admission for these patients, as this may potentially mitigate their short-term mortality risk following craniotomy. In contrast, for patients who require preoperative ventilation, appropriately prolonging preoperative wait time did not significantly affect their short-term mortality risk. This differential effect may reflect the distinct pathophysiological states and management priorities in these two patient populations, where ventilator-dependent patients might benefit from preoperative optimization that outweighs the risks of surgical delay. Overall, our findings provide clinicians with a valuable indicator for assessing surgical risk and optimizing patient management.

Based on these findings, we propose several practical applications for neurosurgical practice. We recommend implementing expedited pathways for non-ventilator-dependent patients with intracranial tumors to minimize the interval between admission and surgery. This approach necessitates streamlined preoperative protocols and improved interdepartmental coordination. When resources are constrained, preoperative wait time should be considered in surgical prioritization decisions, with cases exceeding critical wait time thresholds receiving higher priority given the 8.3% increase in mortality risk per additional day. Healthcare systems should establish quality metrics for acceptable wait times (ideally <1 day for non-ventilator-dependent patients) and incorporate this risk information into patient counseling. Notably, our results suggest differentiated management strategies based on ventilator dependency status: while non-ventilator-dependent patients significantly benefit from prompt intervention, ventilator-dependent patients may allow more flexibility in surgical scheduling to optimize their preoperative condition.

The strengths of our study include the following: Firstly, it involved 18,298 participants, ensuring a substantial sample size and a robust dataset for thorough analysis. Secondly, minimal data on covariates was missing, enabling the adjustment for various confounding factors and the evaluation of multiple model effects. Thirdly, smooth curve fitting was employed to explore the exposure-outcome association. Fourthly, subgroup analyses and interaction tests were conducted to evaluate the consistency of the exposure-outcome relationship across subgroups of confounding factors and to assess the potential effect modification. Fifthly, we performed propensity score matching to control the influence of potential confounding factors and establish patient cohorts with comparable baseline characteristics. Sixthly, we performed extensive sensitivity analyses to validate the robustness of the main results, including categorical analysis, trend testing,

multivariate analysis repetition, and E-value calculation. In summary, this research adhered to a stringent methodology aligned with the STROBE statement, consistent with established practices in the field and presenting reliable results.

The limitations of our study include the following: Firstly, given the retrospective study design and purely associative nature of the results, this study can only establish associations. Consequently, we cannot determine the causal relationship between wait time and thirty-day mortality. Secondly, previous studies have indicated that specific perioperative factors, such as ASA classification [7] and hematocrit [16], correlate with short-term postoperative outcomes. This suggests that prolonged wait time may be merely one of several surrogate predictors for adverse outcomes. Thirdly, since this is a secondary analysis utilizing a public database, it cannot rule out the influence of certain unmeasured confounding factors on the main results (e.g., environment, genetics, pre-hospital wait time, epilepsy, medication status, tumor size and pathology, and intraoperative factors like hypotension, hypercarbia, metabolic acidosis, blood loss and transfusion). However, we calculated the E-value to quantify the potential impact of unmeasured confounders and concluded that they were unlikely to nullify the observed association. Fourthly, the original database lacked direct data on tumor location and tumor type; therefore, the variables 'surgical site' and 'tumor type' were inferred from the CPT codes linked to each surgical procedure. This indirect inference method may be associated with a certain degree of uncertainty and risk of misclassification. Fifthly, the generalizability of the findings may be limited as this study was conducted solely in the United States, where both population characteristics and clinical practices may differ from other regions. Sixthly, certain statistically significant variations in our results might be attributed to the high statistical power afforded by large sample size. For instance, the baseline variations in BMI, Na, and INR across wait time groups are minimal and not clinically important. Seventhly, our primary outcome was thirty-day mortality, which can be influenced by various secondary outcomes—postoperative complications such as infection, hemorrhage, myocardial infarction, pulmonary embolism, stroke, and reoperation. Since this study did not analyze the associations between wait time and these secondary outcomes, we cannot infer the potential mediating factors (some postoperative complications) that might contribute to the observed association between wait time and thirty-day mortality. Eighthly, our findings should be considered exploratory since the analyses are based on a single registry database and have not yet been rigorously validated in an independent external database.

Considering the limitations outlined above, future research should concentrate on the following aspects: Firstly, establishing causal mechanisms between wait time and thirty-day mortality through prospective studies with rigorous designs, particularly analyzing postoperative complications (such as infection, bleeding, and myocardial infarction) as potential mediating factors in this relationship. Secondly, incorporating previously unmeasured confounding factors to minimize the potential impact of residual confounding on the observed association. Thirdly, validating findings through multi-center studies or by utilizing independent external databases to enhance the generalizability and reliability of results. By addressing these aspects, future studies can confirm and expand our findings, ultimately leading to improved patient care and surgical planning in neurosurgical practice.

## Conclusions

This study, focusing on a large U.S. cohort, is the first to identify an independent linear association between preoperative wait time and postoperative thirty-day mortality following intracranial tumor craniotomy in non-ventilator-dependent adult patients. Prolonged wait time was significantly associated with elevated thirty-day mortality. These findings can help to optimize surgical risk assessment and wait time management, guide clinicians to minimize preoperative wait time, and thereby mitigate the risk of postoperative thirty-day mortality. Nonetheless, due to the associative nature of the results and retrospective study design, further research is warranted to validate the results and establish causality.

## Supporting information

**S1 Table. Surgical site, tumor type and corresponding CPT codes.**
(DOCX)

**S2 Table. The univariate analyses of thirty-day mortality.**
(DOCX)

**S3 Table. The comparison of multivariate analysis results from three datasets.**
(DOCX)

**S1 Data. The database made public by Zhang et al and used in our study.**
(CSV)

## Acknowledgments

All authors extend their appreciation to Zhang et al for originally uploading and making public of the data available for our secondary analysis.

## Author contributions

**Data curation:** Weifeng Dong.

**Formal analysis:** Zhichao Gao, Yuhang Zhang.

**Investigation:** Zhichao Gao, Yuhang Zhang, Jiaqing Guan.

**Methodology:** Zhichao Gao.

**Project administration:** cheng huang.

**Supervision:** cheng huang.

**Writing – original draft:** Zhichao Gao, Yuhang Zhang, Jiaqing Guan.

**Writing – review & editing:** Weifeng Dong, cheng huang.

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
