## [Decision Letter · Decision Letter 0]

19 Mar 2025

PONE-D-25-02500Prolonged Preoperative Wait Time Associated with Elevated Postoperative Thirty-Day Mortality Following Intracranial Tumor Craniotomy in Adult Patients : A Retrospective Cohort StudyPLOS ONE

Dear Dr. huang,

Thank you for submitting your manuscript to PLOS ONE. After careful consideration, we feel that it has merit but does not fully meet PLOS ONE’s publication criteria as it currently stands. Therefore, we invite you to submit a revised version of the manuscript that addresses the points raised during the review process.

We look forward to receiving your revised manuscript.

Kind regards,

Barry Kweh

Academic Editor

PLOS ONE

Additional Editor Comments:

An interesting article which requires methodological clarification especially the presentation of the data in tabulated format, as well as justification of the authors use of the Cox regression analysis.

Reviewers' comments:

Reviewer's Responses to Questions

**Comments to the Author**

1. Is the manuscript technically sound, and do the data support the conclusions?

Reviewer #1: Partly

Reviewer #2: Yes

2. Has the statistical analysis been performed appropriately and rigorously? 

Reviewer #1: Yes

Reviewer #2: Yes

3. Have the authors made all data underlying the findings in their manuscript fully available?

Reviewer #1: No

Reviewer #2: Yes

4. Is the manuscript presented in an intelligible fashion and written in standard English?

Reviewer #1: Yes

Reviewer #2: Yes

5. Review Comments to the Author

Reviewer #1: Thank you for the opportunity to review the manuscript regarding “Prolonged Preoperative Wait Time Associated with Elevated Postoperative Thirty-Day Mortality Following Intracranial Tumor Craniotomy in Adult Patients: A Retrospective Cohort Study”. The manuscript provided the vital information for preoperative issue with large sample size. However, there are some issue to be clarify for the improvement and reproducibility.

Introduction

The introduction was quite long (5 paragraphs). Please rewrite the concise introduction and the clear point of the rationale of the study.

Methodology

1. The methodology section was quite brief considering the possibility of reproducibility but the statistical analysis was too long, complex and sophisticate for normal reader (non-statistician) to understand. The author should focus on the detail of the variable selection.

- Since the secondary data came from different institute, what was the definition of each variable? The separate section of the main exposure (the definition, how to determine), outcome of the study (defined the time to event and censored), and potential confounding variables dividing into preop-, intraop-, and postoperative factor should be provided.

- Perioperative factor and complication may affect the outcome of 30 day-mortality after craniotomy. The authors should provide intraoperative adverse event (intraoperative hypotension, hypercarbia, metabolic acidosis), the intraoperative factors (estimated blood loss, blood transfusion, crystalloid volume), and postoperative complications such as bleeding disorder, intracranial hypertension, brain edema, or repeated surgery, that can lead to deteriorate condition postoperatively.

2. Please summary to the statistical analysis section to be simple, concise for non-statistician to understand and describe how to select confounding variable for propensity matching since it is important to include intraoperative and postoperative complication as well.

3. Sample size determination was missing in the manuscript. Considering the large sample size, the small differences can lead to statistically significant difference but not clinically significant. Therefore, it is better to present the proper sample size based on the primary objective.

Results

1. For Cox regression analysis, the cumulative hazard survival curve or Kaplan-Meier curve of 30- day mortality among short and long wait times should be provide to simpler visualize the risk/rate of the 30-day mortality among short and long wait times.

2. Variables in Table 1 was quite confusing. Which variables were postoperative factor (bleeding disorder??), the authors should differentiate the covariates into preop-, intraop-, and postoperative factor since they could be the important factors that lead to mortality after surgery.

Discussion

Please provide the implication of the study in the discussion, so the reader/physician can apply for clinical practice in their hospital setting.

Reviewer #2: Gao Z et al. reported on a clinical study designed to investigate the association between preoperative waiting time and 30-day mortality in intracranial tumor resection. This topic is clinically important and the study is highly significant because shorter waiting times may lead to improved patient outcomes.

The statistical analysis methods were generally appropriate. In addition, many limitations of the study were carefully described in the DISCUSSION. Some minor issues are listed below.

1) Line 160-162: Did the missing data occur for only the seven variables described here? Were there any missing data in categorical data? The authors supplemented medians or means for missing continuous data. However, since single completions should be avoided and the distribution is centered, results based on multiple completions should have been reported as the main analysis and this method should have been positioned in the sensitivity analysis.

2) Line 201-202: Did E value assess the relationship between INR and 30-day mortality? The reason for evaluating against INR rather than preoperative waiting time needs to be carefully explained.

6. PLOS authors have the option to publish the peer review history of their article (what does this mean? ). If published, this will include your full peer review and any attached files.

**Do you want your identity to be public for this peer review?** For information about this choice, including consent withdrawal, please see our Privacy Policy .

Reviewer #1: No

Reviewer #2: No

---

## [Author Response · Author response to Decision Letter 1]

28 Apr 2025

Dear Editor and Reviewer,

We greatly appreciate your valuable comments and suggestions. Below are our responses to each of the comments raised:

Editor Comments An interesting article which requires methodological clarification especially the presentation of the data in tabulated format, as well as justification of the authors use of the Cox regression analysis.

Response:

The data section presented in tabular form in our research may not be sufficiently clear, so we have adjusted the tables and categorized the variables (broadly divided into preoperative factors and intraoperative factors, with preoperative factors further classified into demographic characteristics, clinical features, laboratory indicators, and comorbidities) to present the data more clearly. We have also provided more detailed explanations of the primary exposure, primary outcomes, and covariates in the research methodology section.

The reason we chose the Cox proportional hazards regression model is as follows: This study is a retrospective cohort study, with the primary outcome being 30-day postoperative mortality. The ACS NSQIP database provides detailed records of the survival days for each patient within the 30-day postoperative follow-up period. Considering the study design and data characteristics, we selected the Cox proportional hazards regression model for analysis. This model incorporates time-to-event information and is particularly suited for handling survival time data, which is critical for accurately estimating hazard ratios and understanding the temporal dynamics of mortality risk. We have provided a brief explanation of this in the statistical analysis section of the revised manuscript.

Reviewer #1

Comment #1�The introduction was quite long (5 paragraphs). Please rewrite the concise introduction and the clear point of the rationale of the study.

Response:

Thank you for your valuable suggestion regarding our introduction section. We fully agree that the introduction should be concise and clearly highlight the rationale and purpose of the study.

Following your recommendation, we have comprehensively revised the introduction:

Reduced Length: We have condensed the original 5-paragraph introduction to 4 paragraphs, reducing the overall word count by approximately 30% for a more focused presentation.

Optimized Structure: We have reorganized the content to establish a clearer logical flow:

First paragraph: Directly introduces the epidemiology and clinical significance of intracranial tumors, emphasizing the importance of thirty-day mortality rates.

Second paragraph: Systematically summarizes known risk factors while highlighting that current models overlook preoperative wait time as a potentially modifiable factor affecting patient outcomes.

Third paragraph: Focuses on the evidence regarding preoperative wait time as a potential risk factor and identifies the knowledge gap.

Fourth paragraph: Clearly articulates the research objective and potential clinical value.

The revised introduction now focuses more sharply on the core issues, eliminates redundancy, while retaining the key information supporting the study rationale. We believe these modifications have resulted in a more concise and powerful introduction that better guides readers to understand the value and necessity of our research.

Comment #2�The methodology section was quite brief considering the possibility of reproducibility but the statistical analysis was too long, complex and sophisticate for normal reader (non-statistician) to understand. The author should focus on the detail of the variable selection.

- Since the secondary data came from different institute, what was the definition of each variable? The separate section of the main exposure (the definition, how to determine), outcome of the study (defined the time to event and censored), and potential confounding variables dividing into preop-, intraop-, and postoperative factor should be provided.

- Perioperative factor and complication may affect the outcome of 30 day-mortality after craniotomy. The authors should provide intraoperative adverse event (intraoperative hypotension, hypercarbia, metabolic acidosis), the intraoperative factors (estimated blood loss, blood transfusion, crystalloid volume), and postoperative complications such as bleeding disorder, intracranial hypertension, brain edema, or repeated surgery, that can lead to deteriorate condition postoperatively.

Comment #3�Please summary to the statistical analysis section to be simple, concise for non-statistician to understand and describe how to select confounding variable for propensity matching since it is important to include intraoperative and postoperative complication as well.

Response:

Thank you for your valuable methodological suggestions. We understand the balancing challenge in your recommendations: ensuring research reproducibility and clarity of variable definitions while keeping statistical analysis concise and understandable. In response to your suggestions, we have comprehensively revised the methodology section:

(1).Variable Section Restructuring and Enhancement:

Primary Exposure: Clearly defined preoperative wait time as "the interval (in days) from hospital admission to surgical intervention" and explained how it is recorded in the ACS NSQIP database, as well as how it was analyzed both as a continuous and categorical variable.

Primary Outcome: Clarified that thirty-day mortality refers to "all-cause death occurring within 30 days after the surgical procedure," and detailed how this outcome was determined and followed up in the database.

Covariates: Systematically categorized all variables into preoperative and intraoperative factors, improving structural clarity.

(2).Statistical Analysis Optimization: We maintained necessary statistical details to ensure research reproducibility while improving readability through:

Structured Presentation: Reorganizing the statistical analysis section into a logically clear sequence of steps

Explaining Confounding Variable Selection Criteria: Clearly specified that confounding variables were selected based on three aspects: clinical experience, literature reports, and statistical results (statistical significance and impact on the primary relationship).

Explaining Statistical Method Selection: Clearly articulated the rationale for statistical method selection, particularly the use of Cox proportional hazards models

Propensity Score Matching: Clarified that all previously identified confounding variables were used for matching to ensure comparability between groups

(3).Regarding Intraoperative and Postoperative Variables:

Intraoperative Factors: We included all available intraoperative factors from the database (tumor type, surgical site, operation time, emergency cases, etc.). Unfortunately, the specific intraoperative parameters you mentioned (intraoperative hypotension, blood loss, etc.) were not available in the ACS NSQIP database, and we have discussed these unmeasured confounding factors in the limitations of our study.

Regarding Postoperative Complications: From a methodological perspective, postoperative complications (available in the ACS NSQIP database, including infection, bleeding, myocardial infarction, stroke, etc.) lie on the causal pathway between preoperative wait time and postoperative death, making them potential mediators rather than confounders. Including mediators in multivariable analysis and propensity score matching would lead to over-adjustment bias and underestimate the total effect of wait time on mortality. Instead, we have acknowledged in the limitations section of our discussion that not analyzing these potential mediating factors is a limitation of our study, and we suggest future research explore these mediating mechanisms. This approach aligns with modern epidemiological research methods.

We believe this revised approach complies with STROBE guidelines for research transparency and reproducibility while improving the overall readability of the methodology section. Through reorganization and clearer explanations, we have enhanced accessibility for readers without statistical expertise while maintaining the integrity of our research.

Comment #4�Sample size determination was missing in the manuscript. Considering the large sample size, the small differences can lead to statistically significant difference but not clinically significant. Therefore, it is better to present the proper sample size based on the primary objective.

Response:

Thank you for your valuable suggestion regarding sample size determination. Concerning this issue, we would like to clarify the following points:

First, as this is a retrospective cohort study, sample size was determined through a systematic filtering process from the original database using clearly defined inclusion and exclusion criteria, rather than being pre-calculated. We have provided a detailed screening flowchart of the study population in the methods section. In retrospective research designs, artificially reducing sample size could potentially introduce selection bias, which contradicts the purpose of the study. Pre-calculation of sample size is primarily applicable to randomized controlled trials and prospective study designs.

Second, we fully understand the reviewer's concern that "large sample sizes may render clinically insignificant small differences statistically significant." In fact, we have already acknowledged this in the limitations section of our discussion: "Certain statistically significant variations in our results might be attributed to the high statistical power afforded by large sample size. For instance, the baseline variations in BMI, Na, and INR across wait time groups are minimal and not clinically important."

Third, to address this challenge, our analysis not only relied on p-values but also focused on reporting effect sizes, including odds ratios (OR) and hazard ratios (HR) with their 95% confidence intervals, allowing readers to directly assess the clinical significance of the effects. More importantly, we conducted multiple sensitivity analyses to verify the stability of our main findings, including categorical analysis, trend tests, propensity score matching, and E-value calculations. These methods collectively ensured that our discoveries have not only statistical significance but also clinical practical value.

In conclusion, while large sample studies indeed face potential discrepancies between statistical significance and clinical significance, we have thoroughly considered this challenge in our study design and results interpretation, and have employed appropriate statistical methods and careful result interpretation to ensure the scientific value of our research findings.

Comment #5�For Cox regression analysis, the cumulative hazard survival curve or Kaplan-Meier curve of 30- day mortality among short and long wait times should be provide to simpler visualize the risk/rate of the 30-day mortality among short and long wait times.

Response:

Thank you for your valuable suggestion. We completely agree that providing Kaplan-Meier curves would allow for more intuitive visualization of the differences in 30-day mortality among different wait time groups.

In response to your suggestion, we have added the following content to our revised manuscript:

In the methods section, we added "Kaplan-Meier analysis: The effects of wait time categories on thirty-day mortality were evaluated using Kaplan-Meier curves generated with the log-rank test."

In the results section, we included the corresponding analysis results: "Fig 3 displays the results of Kaplan–Meier analysis, demonstrating a significantly higher overall cumulative hazard of thirty-day mortality in patients with longer wait times (1-7 day and >7 days) compared to those wait less than one day (P<0.001)."

We have added Fig 3 to visually demonstrate the differences in mortality risk across different wait time groups.

We believe this additional visualization will help readers more clearly understand the differences in postoperative mortality risk among patients with different waiting times, making our research findings more intuitive and clear.

Comment #6�Variables in Table 1 was quite confusing. Which variables were postoperative factor (bleeding disorder??), the authors should differentiate the covariates into preop-, intraop-, and postoperative factor since they could be the important factors that lead to mortality after surgery.

Response:

Thank you for your valuable suggestion regarding variable classification. The issue you raised is very important, and we have reorganized and clearly categorized the variables in Table 1:

We have systematically divided all covariates into preoperative and intraoperative factors, with clear labeling in Table 1:

Preoperative Factors include:

1.Demographic characteristics: sex, age ranges, race, and smoking status.

2.Clinical characteristics: BMI, ventilator dependent, functional health status, steroid use for chronic condition, preoperative blood transfusion, and ASA classification.

3.Laboratory indicators: serum sodium, blood urea nitrogen, white blood cell counts, hematocrit, and international normalized ratio.

4.Comorbidities: diabetes, hypertension, severe COPD, congestive heart failure, renal failure/dialysis, disseminated cancer, open wound infection, preoperative systemic infection, and bleeding disorders.

Intraoperative Factors include: tumor type, surgical site, operation time, emergency case, and wound classification.

It's important to note that "bleeding disorders" mentioned by the reviewer is classified in our study as a preoperative comorbidity, referring to pre-existing bleeding tendency conditions (such as hemophilia, platelet function disorders, etc.) that patients had before surgery, rather than postoperative bleeding complications.

Regarding postoperative factors, as explained in our previous response, our methodological design intentionally focused the analysis on preoperative and intraoperative factors to avoid over-adjustment bias that would result from including potential mediator variables.

Through this systematic reorganization and clear categorization of variables, we believe Table 1 now more clearly demonstrates the baseline characteristic differences between groups.

Comment #7�Please provide the implication of the study in the discussion, so the reader/physician can apply for clinical practice in their hospital setting.

Response:

Thank you for your valuable suggestion regarding the clinical implications of our study. We completely agree that providing practical applications is crucial for readers/physicians to implement in their hospital settings.

In response to your suggestion, we have added a dedicated paragraph in the discussion section that specifically elaborates on the clinical applications of our findings:

"Based on these findings, we propose several practical applications for neurosurgical practice. We recommend implementing expedited pathways for non-ventilator-dependent patients with intracranial tumors to minimize the interval between admission and surgery. This approach necessitates streamlined preoperative protocols and improved interdepartmental coordination. When resources are constrained, preoperative wait time should be considered in surgical prioritization decisions, with cases exceeding critical wait time thresholds receiving higher priority given the 8.3% increase in mortality risk per additional day. Healthcare systems should establish quality metrics for acceptable wait times (ideally <1 day for non-ventilator-dependent patients) and incorporate this risk information into patient counseling. Notably, our results suggest differentiated management strategies based on ventilator dependency status: while non-ventilator-dependent patients significantly benefit from prompt intervention, ventilator-dependent patients may allow more flexibility in surgical scheduling to optimize their preoperative condition."

Additionally, we expanded the explanation of the differential effects between ventilator-dependent and non-ventilator-dependent patients, clarifying that this differential effect may reflect the distinct pathophysiological s

---

## [Editor Report · Decision Letter 1]

4 May 2025

Prolonged Preoperative Wait Time Associated with Elevated Postoperative Thirty-Day Mortality Following Intracranial Tumor Craniotomy in Adult Patients : A Retrospective Cohort Study

PONE-D-25-02500R1

Dear Dr. Huang,

We’re pleased to inform you that your manuscript has been judged scientifically suitable for publication and will be formally accepted for publication once it meets all outstanding technical requirements.

Kind regards,

Barry Kweh

Academic Editor

PLOS ONE

Additional Editor Comments (optional):

The authors have satisfactorily addressed methodological and statistical concerns regarding their findings, ensured the introduction is more concise and broadened the discussion regarding other similar outcomes and mortality following tumour surgery.
---

## [Editor Report · Acceptance letter]

PONE-D-25-02500R1

PLOS ONE

Dear Dr. huang,

I'm pleased to inform you that your manuscript has been deemed suitable for publication in PLOS ONE. Congratulations! Your manuscript is now being handed over to our production team.

Kind regards,

on behalf of

Dr. Barry Kweh

Academic Editor

PLOS ONE